# The association between upper gastrointestinal endoscopic findings and internal radiation exposure in residents living in areas affected by the Chernobyl nuclear accident

Yesbol Sartayev[1,2], Izumi Yamaguchi[3], Jumpei Takahashi[4], Alexander Gutevich[5], Naomi Hayashida[1,2]*

1 Life Sciences and Radiation Research, Nagasaki University Graduate School of Biomedical Sciences, Nagasaki, Japan, 2 Division of Strategic Collaborative Research, Atomic Bomb Disease Institute, Nagasaki, Japan, 3 Ueno Hospital, Fukuoka, Japan, 4 Center for International Collaborative Research, Nagasaki University, Nagasaki, Japan, 5 Zhytomyr Inter-Area Medical Diagnostic Center, Korosten, Ukraine

* naomin@nagasaki-u.ac.jp

**Data Availability Statement:** All relevant data are within the manuscript and its Supporting information files. However, researchers who use

## Abstract

Many people living around the Chernobyl Nuclear Power Plant (CNPP) have been exposed to $^{137}$Cs for several decades after the CNPP accident. Although half-life of $^{137}$Cs is about 30 years, some wild forest foodstuffs are contaminated by $^{137}$Cs even now. We pointed out in a previous report that low-dose internal radiation has been occasionally detected in people's body. Moreover, some doctors in local hospitals have claimed that internal exposure from contaminated foodstuffs may affect the digestive organs and possibly cause gastrointestinal (GI) diseases. Thus, we attempt to assess whether internal radiation exposure affects digestive organs or not, and the possible factors that influence digestive organs. Overall, 1,612 residents were assessed for internal $^{137}$Cs concentration using Whole-Body Counter and their digestive organs were screened with upper GI endoscopy from 2016–2018 in the Zhytomyr region, Ukraine. All participants answered to the questionnaire including their background, intake of wild forest foodstuff, intake frequency, smoking habits, and alcohol consumption. We checked the number of upper GI endoscopic diagnosis per person to assess the extent of damage to the upper digestive organs. Next, we statistically analyzed associations between this number and age, sex, level of internal exposure dose, alcohol consumption, wild forest foodstuff intake, and smoking. Consequently, we revealed that the number of GI diagnosis is significantly increased by factors such as sex, intake of wild forest foodstuff, and alcohol consumption. However, the average level of internal exposure of $^{137}$Cs and smoking did not relate to the number of GI diagnosis. Thus, the results of multiple regression revealed that alcohol consumption is independently related to the number of GI diagnosis that is most likely accompanied by the intake of wild forest foodstuff. In conclusion, the low-dose internal exposure may not affect the digestive organs of residents living around CNPP.

the data in their research must get permission from the research ethics review board at the organization of their affiliationand the Ethics Committee at Nagasaki University Graduate School of Biomedical Sciences. The contact information for the ethics committee is as follows: Tel.: +81–95–819–7198 Website: http://www.mdp.nagasaki-u.ac.jp/research/support_rinri.html.

**Funding:** This research was supported by financial support from the Program of the Network-type Joint Usage/Research Center for Radiation Disaster Medical Science (URL: https://housai.hiroshima-u.ac.jp/en/) and the Atomic Bomb Disease Institute of Nagasaki University (URL: https://www.genken.nagasaki-u.ac.jp/abdi/index.html). The funders had no role in study design, data collection and analysis, decision to publish, or preparation of the manuscript.

**Competing interests:** The authors have declared that no competing interests exist.

## Introduction

The Chernobyl Nuclear Power Plant (CNPP) accident occurred in Ukraine in 1986 during a safety test in the steam turbine of a nuclear reactor. Two employees died immediately at the time of the accident, and 28 more firefighters died within several weeks after receiving lethal doses of ionizing radiation in a brief period. Shortly after the accident, it was claimed that over 50,000 people would die of Chernobyl-induced cancer and some other diseases related to radiation [1]. However, epidemiological studies on the consequences of the Chernobyl accident and its effects on health have reported a considerably smaller number of casualties. In the early 2000s, several prominent global organizations published multiple reports on the effects of low dose radiation on health from the Chernobyl accident. The United States National Research Council published its comprehensive BEIR-VII report dedicated to the effects of low ionizing radiation levels in 2006 [2], along with a series of independent reports from the World Health Organization (WHO), International Atomic Energy Agency (IAEA), and United Nations Scientific Committee on the Effects of Atomic Radiation (UNSCEAR) on the effects of radiation and its consequences for health after the fallout from the Chernobyl accident [3–5]. The BEIR-VII 2006 report concluded, "At this time [2006], no conclusion can be drawn concerning the presence or absence of a radiation-related excess of cancer—particularly leukemia—among Chernobyl accident recovery workers" [2].

Nevertheless, all researchers have agreed that the increase in the rates of thyroid cancer among children under the age of 18 from the affected areas was the consequence of exposure from the fallout of the CNPP [6]. By 2015, more than 20,000 thyroid cancer cases were diagnosed among two million highly contaminated children, who were under 18 at the time of the accident; 15 of them had lethal outcomes [6].

Most epidemiological studies on the effects of radiation on the human body were primarily focused on cancer [7]. Majority of these studies were conducted based on survivors of atomic bombings, participants of radiotherapy procedures, or radiation workers. Radiation effects expressed as non-cancer diseases have been less systematically studied and reported [7]. UNSCEAR implemented major reviews on non-cancer radiation diseases in the 1982 [8] and 1993 [9] UNSCEAR Reports.

The threshold for non-cancer diseases had been considered at dose levels over 4 to 5 Gy, until the Life Span Study (LSS) demonstrated evidence at doses lower than the these [7]. In 1992, the analysis of non-cancer diseases based on mortality data from the LSS cohort and survivors of the atomic bombing in Japan demonstrated a statistically significant association between radiation doses and non-cancer diseases [10]. Furthermore, excessively high risks for mortality from strokes, heart diseases, and respiratory and digestive system diseases were reported.

Analyses of the same LSS non-cancer mortality data from 1965 to 1997 showed little evidence of excess risks below 0.5 Sv. These findings were true for the four major disease categories considered: strokes, coronary heart diseases, and digestive and respiratory diseases [7]. Likewise, Ozasa et al., in their LSS study of victims of atomic bombing conducted from 1950 to 2003, found that the risk of liver cirrhosis and major digestive diseases did not show any increased radiation risks during the whole period or for the period after 1965 (Excessive Relative Risk [ERR]/Gy = 0.11, 95% confidence interval [CI]: -0.07, 0.34 and 0.17, 95% CI: -0.04, 0.42) [11].

There is significant uncertainty about the shape of the dose-response relationship at low doses. The current estimates for lifetime risks are presented for exposure at 1 Sv, where they are barely affected by the shape of the dose-response. The magnitude of the risk of non-cancer diseases at lower dose levels (for example, 0.5 Sv) is ambivalent [7]. Ozasa et al., in their 2012

study of the mortality of Atomic Bomb Survivors, pointed out that the increased risks of non-neoplastic diseases, including the circulatory, respiratory, and digestive systems were observed; however, whether these were causal relationships required further investigation [11].

Some studies showed evidence of the effects of radiation on GI diseases. However, all of them only investigated the effects of external radiation, mostly finding evidence in cohorts with high dose irradiation. Therefore, their findings are not necessarily comparable with ours because our study focuses on internal exposure at relatively low doses that originates from the intake of foodstuff contaminated by $^{137}$Cs in affected areas around Chernobyl. Moreover, until recently, there have been no studies dedicated to studying the effects of internal radiation on digestive organs.

A few studies investigating the presence of radioactivity in the bodies of people living in contaminated areas around CNPP, showed that a substantial percentage of the population—almost 50% in the beginning of the study year—had some level of radiation [12, 13]. A study conducted from 1996 to 2008 found that 513 participants or 0.35% of the study population had an annual internal radiation dose exceeding 1 mSv, which is the dose limit set by the International Commission on Radiological Protection for the general public [12]. Despite our study was conducted from 2009 to 2018, on screening the residents around CNPP for internal radiation, we found fewer residents (53 participants, 0.02%) with higher levels of dose and radiation detected in their bodies. Consequently, there is still uncertainty regarding the effects of chronic low-dose internal radiation and its health outcomes [13].

Some doctors working in clinics in contaminated areas claim and suspect that internal radiation deteriorates the functions of stomach and may adversely affect the GI system of the human body. Therefore, we attempt to assess whether low-dose internal radiation exposure affects digestive organs, and the possible factors influencing the digestive organs of residents around CNPP. The findings of upper GI endoscopic examination were interpreted according to the International Classification of Diseases (ICD) of World Health Organization (WHO). Common upper GI endoscopic diagnoses such as gastritis, duodenitis, duodenogastric reflux, gastroesophageal reflux, stomach ulcer, diaphragmatic hernia, and other diagnoses, including a few cases of cancer, were diagnosed in a wide range of combinations among most of the screened participants. Our study identifies a possible association between the detected number of upper GI endoscopic diagnosis and low-dose internal exposure in participants living in areas contaminated by the Chernobyl accident. This study is one of the first to conduct an investigation of the effects of internal radiation exposure on the GI organs.

## Materials and methods

We conducted this study from July 2016 to February 2018 at the Medical Center of Korosten city. The participants were residents of Korosten city and eight subordinated districts of the Zhytomyr region in Ukraine. As of January 1, 2019, the population of the study area was approximately 323,000. Korosten city is the largest settlement, with over 63,000 people. The study area is located to the west of the CNPP, and the fallout from the accident significantly contaminated it. Our study included the settlements that were between 40–150 kilometers west from the nuclear power plant.

In total, 1,612 people residing within the research area participated in this study. All of them were under the health care surveillance of the Zhytomyr Inter-Area Medical Diagnostic Center (Medical Center) that provides health care services to the residents our research area. We invited all patients who sought medical assistance in the Medical Center for any upper GI symptoms or digestive organ disorders that required GI endoscopic intervention during the study period. Residency registration within the research area at the moment of examination

was mandatory. Those who agreed to participate in the study, initially received a detailed description of the process and content of the study and were then asked to provide written consent. Afterward, they first completed a questionnaire regarding their lifestyle and dietary habits. Questionnaires were distributed in hard copies and prepared in the Russian language. It consisted of four pages in A4 size and included the respondent's name, address, date of birth, informed consent, milk and forest food intake, alcohol consumption, smoking habits and their frequency. Once they finished filling the questionnaire, they were invited to udergo upper GI endoscopy and measurement of their internal body burden on the Whole-Body Counter (WBC). All the data collected from the questionnaires, GI endoscopic examinations, and WBC measurements were then used to assess their effects of internal exposure on the upper GI endoscopic findings and identify their associations and contributions.

To assess the internal exposure dose of participants, we used a WBC manufactured by Aloka Co., Ltd (Japan), equipped with a 7.6 cm diameter NaI (TI) detector. This WBC has an adjustable seat for height and angle so that the examinee can place their abdomen on the detector. The minimum detectable radioactivity level of $^{137}$Cs on this WBC was 270 Bq per body. As for the upper GI examination, all endoscopy screenings were implemented on professional GI endoscopy equipment made by the OLYMPUS Company (Japan). Upper GI endoscopic findings in participants were diagnosed by professional gastroenterologists according to the ICD of WHO. We used the number of upper GI endoscopic diagnosis per person to indicate the extent of GI damages. The number of upper GI endoscopic diagnosis detected in one participant in our dataset varied from 0 to 5 diagnoses. Each participant could have various combinations of upper GI diseases, such as gastritis, duodenitis, duodenogastric reflux, gastro-esophageal reflux, stomach ulcer, diaphragmatic hernia, and so on. We considered the number of upper GI endoscopic diagnosis detected in one person and assessed the effect of internal exposure and other factors on the increase of upper GI disease. All measurement procedures and endoscopy screenings were performed by qualified medical personnel at the Medical Center. Measured levels of radioactivity in the body and the number of detected upper GI endoscopic diagnosis in each participants were first written down on hard copies of registry cards of each participant by medical specialists. Thereafter, they were transferred into the Excel format, along with the appropriate information from the questionnaire for further statistical analysis on professional software.

All the data was cleaned, filtered and grouped by certain characteristics, such as sex, age, number of detected upper GI endoscopic diagnosis, wild forest food and alcohol intake, and smoking habits. We also converted Bq/body from the WBC into Bq/kg for each individual and subsequently stratified them into two groups—participants with detectable level of $^{137}$Cs and participants with non-detectable level of radioactivity. When the internal exposure of the participants was below detectable levels, they were considered and qualified as "0 Bq." The relevant and necessary statistical tests were conducted and represented in the appropriate way. All statistical analyses were performed on IBM SPSS Statistics 25.0 software. The Mann-Whitney U test and Chi-square tests were used for statistical significance and the determination of averages and proportions. We also ran correlation tests and univariate regression analysis to test the contributions of several variables. P-values lower than 0.05 were considered significant.

Following the *"Ethical Guidelines for Medical and Health Research Involving Human Subjects"* published by the Ministry of Education, Culture, Sports, Science, and Technology and the Ministry of Health, Labor and Welfare, this study was approved by the Ethics Committee at Nagasaki University Graduate School of Biomedical Sciences (approval no.: 16062493–4) on March 31, 2021. Informed consent was obtained from each individual through a written form, that indicated agreement for participation in the research. All relevant data excluding the personal information of patients in this research are available upon request.

## Inclusivity in global research

Additional information regarding the ethical, cultural, and scientific considerations specific to inclusivity in global research is included in the Supporting Information (S1 Questionnaire).

## Results

About 1,612 participants took part in the study and underwent the WBC screening and GI endoscopy. The general information of the participants is presented in Table 1. Among the participants, 36% were men, and 64% were women. The average age of the participants was 49 years, although the average age of the female group was (51 years), significantly higher than that of men (46 years) (p<0.001). The number of upper GI endoscopic diagnosis detected in one person ranged from 0–5 diagnoses. Almost all participants had some type of diagnosis, except for 16 participants, who had "0" diagnosis and were classified as "healthy." Among these healthy participants, two were male, and the rest were female. The average number of upper GI endoscopic diagnosis for the entire study population was two, though men had a significantly higher average number (2.2±0.8) than women (1.9±0.8).

Table 2 represents the average levels of internal radiation from $^{137}$Cs for the entire population, and individually for men and women. The average Bq/kg of internal exposure detected in all participants was 6.2±11.8 Bq/kg. Although the average level of internal radiation in men was higher compared to women, it is not significantly higher (6.7 Bq/kg) than women's average internal exposure (5.9 Bq/kg) (p = 0.182). Similarly, men had a higher proportion of those with detectable levels (33%) than in women (29%), again with no significant difference (p = 0.067).

Fig 1A illustrates the distribution of the age ranges of the whole population in numbers and proportion. The 81–83 years age range was the smallest with only 6 individuals and accounted for less than 0.1%, followed by the age range of 18–20 years, accounting for 3% of the entire population. The largest age ranges by number of participants were 51–60 years and 61–70

**Table 1. General information of participants and number of upper GI endoscopic diagnosis.**

|  | All | Men | Women | P-value |
|---|---|---|---|---|
| **Total participants, (%)** | 1612 | 581 (36.0) | 1031 (64.0) | |
| **Avg. age, n±SD** | 49±16.2 | 46±16.2 | 51±15.9 | < 0.001 |
| **Avg. number diagnosis** | 2.0±0.8 | 2.2±0.8 | 1.9±0.8 | < 0.001 |
| **Positive diagnosis, n. (%)** | 1596 | 579 (99.7) | 1017 (98.6) | 0.049 |
| **Negative diagnosis, n. (%)** | 16 | 2 (0.3) | 14 (1.4) | |

P-value represents the significance in value of men and women.

**Table 2. Body burden and participants with detected radioactivity.**

|  | All | Men | Women | P-value |
|---|---|---|---|---|
| **Avg. Bq/kg, n±SD (All participants)** | 6.2±11.8 | 6.7±10.9 | 5.9±12.2 | 0.182 |
| **Number of detected particiapants, n. (%)** | 485 (30) | 191 (33) | 294 (29) | 0.067 |
| **Avg. Bq/kg of detected participants, n±SD** | 20.6±12.8 | 20.4±8.9 | 20.7±14.8 | 0.828 |

Body burden is the value of radioactivity in participants represented in Bq/kg.

Detected participants are those participants who have any value of radioactivity above the minimum detectable radioactivity.

P-value represents the significance in values of men and women.

(A)

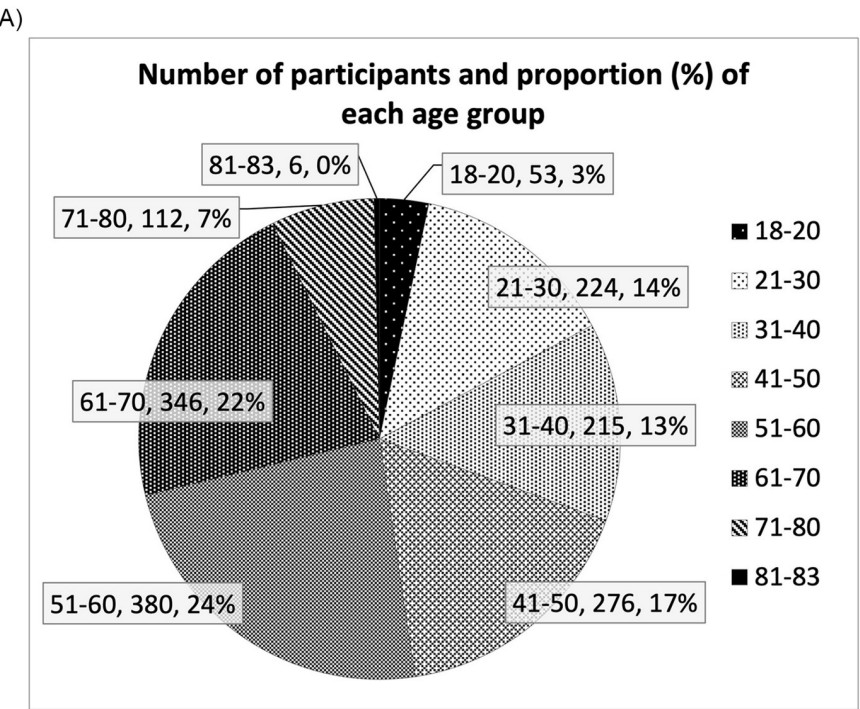

(B)

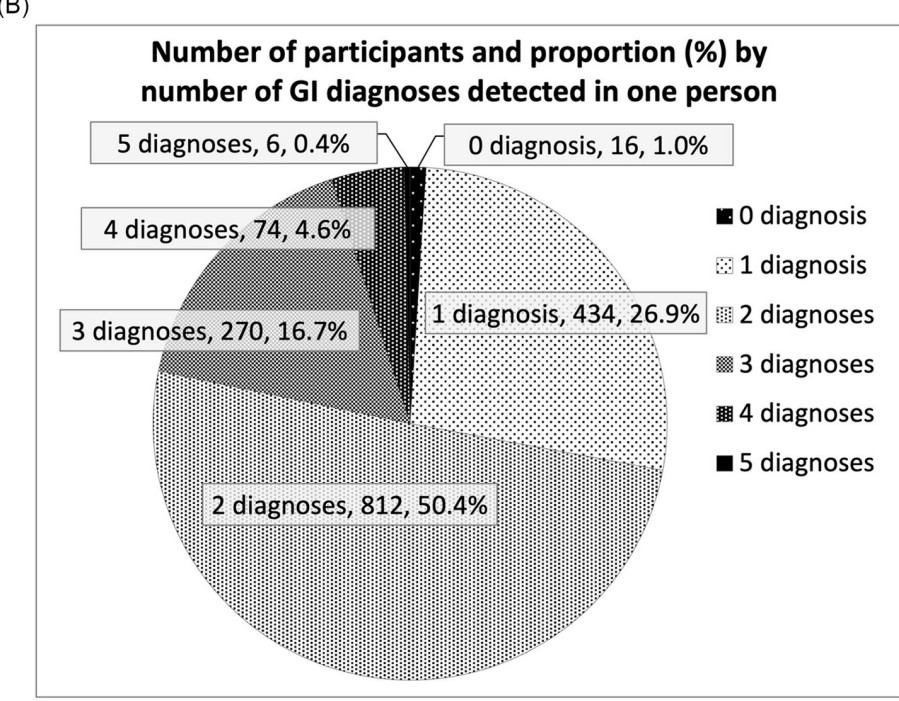

**Fig 1. Background of the participants.** (A) The actual number and percentage of each age range in the entire population. Most major age ranges are distributed relatively evenly, except the youngest and oldest age groups, consisting of 3% and less than 0.1% respectively, of the entire population. (B) The amount and percentage of each number of upper GI endoscopic diagnosis detected in one person. The most prevalent numbers of upper GI endoscopic diagnosis were 1, 2, and 3 diagnoses, which comprised 94%. The most prevalent number of upper GI endoscopic diagnosis indicated in one person was 2, comprising 50% of the total population.

years, comprising 24% and 22% respectively. We also analyzed the distribution of the numbers of upper GI endoscopic diagnosis detected in one individual (Fig 1B). It shows the number and proportion of participants categorized by the number of upper GI endoscopic diagnosis detected in one person. In our data set, the number of upper GI endoscopic diagnosis detected in an individual ranged from 0–5. The prevalent number of upper GI endoscopic diagnosis detected in an individual was two, accounting for 50%, and the least prevalent was five, accounting for less than 0.1%. Fig 1B shows that more than 70% of the study population had two or more diagnoses. The types of upper GI findings varied to a vast extent. However, all of them were commonly prevalent in the general population, except eight rare cases of cancers (six stomach cancers and two esophagus cancers). The most frequently detected upper GI endoscopic diagnosis throughout our study were gastritis, duodenitis, duodenogastric reflux, gastroesophageal reflux, stomach ulcer, diaphragmatic hernia, and other rare diagnosis. However, we did not investigate the characteristics of upper GI endoscopic diagnosis as a part of this study.

Fig 2 shows the average Bq/kg and the proportion of number of the upper GI endoscopic diagnosis in each age range group. Young people aged 18–20 years were dominantly detected for 1 and 2 diagnoses. However, the proportion of number of upper GI endoscopic diagnosis increased with the increasing age of the groups. This tendency can be seen in the proportions of 3 and 4 diagnoses. However, the proportion of 5 diagnoses was detected only in two age groups, namely 51–60 years and 61–70 years. We also assessed the significance of the average Bq/kg between age groups using ANOVA test. The ANOVA multiple Comparisons Tamhane test showed no significant difference between age groups (p = 0.461). The highest average Bq/kg was observed in the group of 81–83 years old. However, this group may not be representative, as it only contains 6 participants, while all other groups consist of 53–380 participants.

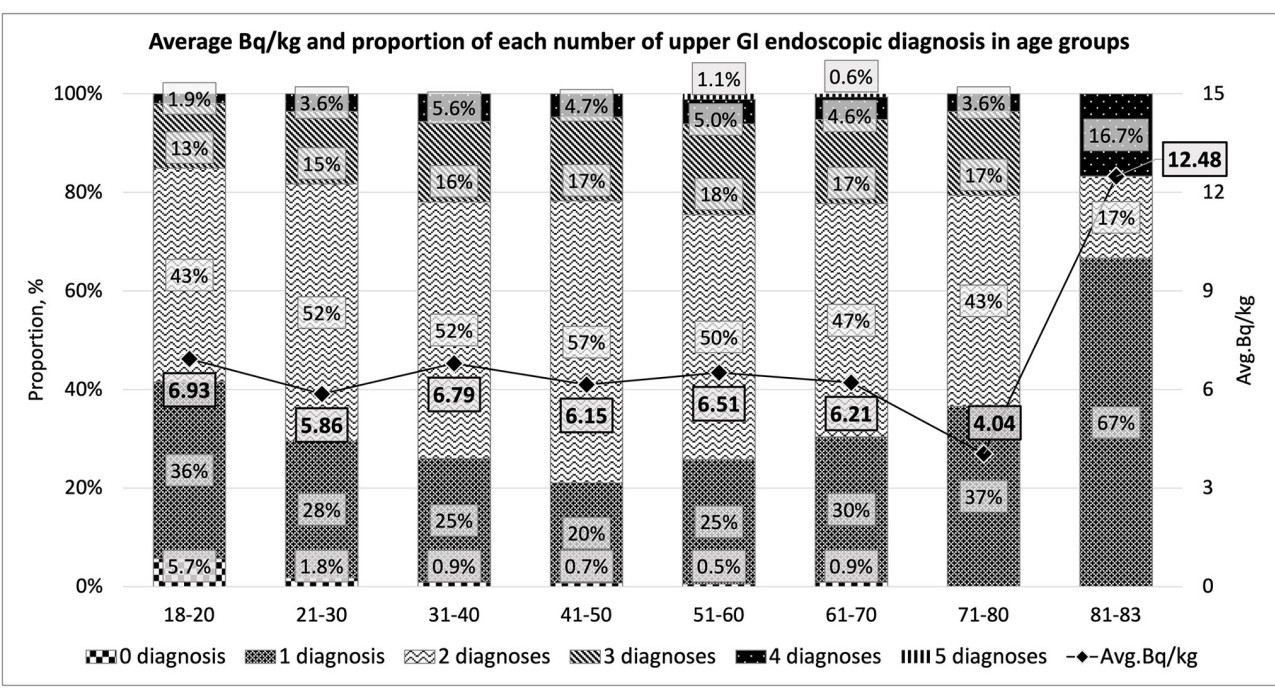

**Fig 2. Average Bq/kg and proportion of number of upper GI endoscopic diagnosis in each age group.** The stretched line throughout all age ranges indicates the average Bq/kg detected in respective age groups. The number of upper GI endoscopic diagnosis and proportions increased with age, although the average Bq/kg was not indicative.

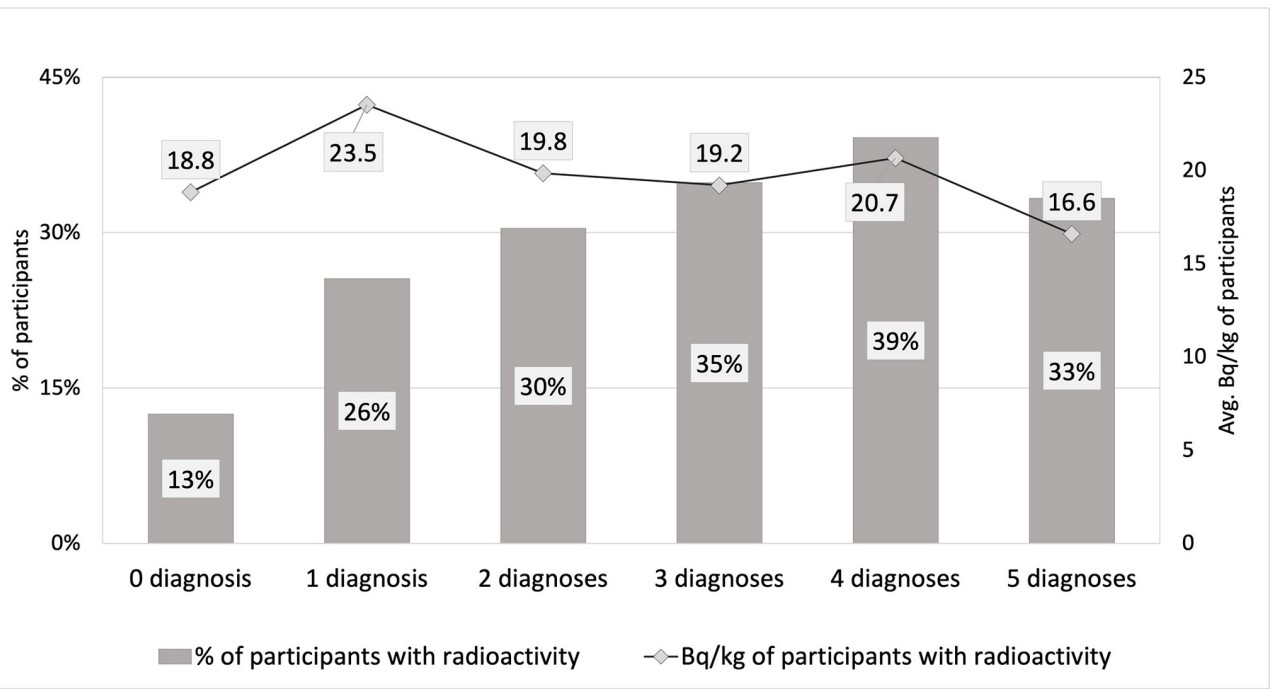

**Fig 3. Bq/kg and percentage of participants with detected radiation in each number of upper GI endoscopic diagnosis groups. Avg**. Despite the percentage of detected people seemingly increasing with the increase in the number of upper GI endoscopic diagnosis, the black line shows the average Bq/kg and implies no association with the same.

If the age group of 81–83 years is excluded, then the younger the age, the higher the level of internal radiation exposure. The middle and elderly age groups had relatively lower exposure to $^{137}$Cs.

Fig 3 demonstrates the average Bq/kg and percentage of participants with $^{137}$Cs exposure above the WBC's detection limit for each number of upper GI endoscopic diagnosis in the group. It shows the possibility of a relationship between the number of upper GI endoscopic diagnosis and percentage of participants with detectable internal exposure. The percentage of people with detectable radiation increased with the increasing number of upper GI endoscopic diagnosis groups, though the percentage of people with detectable radiation decreased notably in the last group. The correlation coefficient between the average Bq/kg and each number of the upper GI endoscopic diagnosis group was 0.038 (Pearson correlation, p>0.05), suggesting almost no correlation and no significance.

Table 3 shows the results of univariate regression analysis in which the number of upper GI endoscopic diagnosis was a dependent variable. In contrast, age, sex, Bq/kg, wild forest food intake, alcohol consumption, and smoking were independent variables. This analysis evaluates whether these independent variables affect the increase in the number of upper GI endoscopic diagnosis in individuals. It also shows the level of contribution of each factor and its significance. Table 3 represents the results of the regression where factors such age, sex, intake of wild food, and alcohol consumption contribute significantly to the increase in the number of upper GI endoscopic diagnosis. Factors such as Bq/kg and smoking did not affect the increase in the number of upper GI endoscopic diagnosis. The adjusted $R^2$ for this regression analysis was 0.050.

We also performed regression analysis for women separately, as it could reveal an interaction between alcohol consumption and wild food intake, assuming that alcohol consumption

**Table 3. Multiple regression analysis***.

| Independent variables | Unstandardized coefficients (B) | P-value |
|---|---|---|
| **Age** | 0.003 | 0.040 |
| **Sex (Men 1, Women 2)** | -0.307 | 0.001 |
| **Alcohol (No-0, Yes-1)** | 0.148 | 0.001 |
| **Smoking (No-0, Yes-1)** | 0.014 | 0.789 |
| **Bq/kg per person** | 0.002 | 0.311 |
| **Intake of wild food (No-0, Yes-1)** | 0.182 | 0.003 |

*Adjusted R Square = 0.050

is lower in women compared to men. Regression analysis for the women-only group revealed results similar to Table 3, indicating significance for age, alcohol consumption, intake of wild food ($p < 0.05$). However, smoking and the level of Bq/kg seemingly did not affect the number of upper GI endoscopic diagnosis ($p > 0.05$).

## Discussion

In this study, the key focus was on internal exposure emitted from [137]Cs and its potential effects on the GI system of the human body. A majority of studies concerning the CNPP accident that investigate the effects of radiation on the human body have concentrated their primary interests in [137]Cs and internal exposure. [137]Cs has a greater effect on people than other radionuclides due to its properties, such as a long half-life, the amount released into the environment and dispersion in a wider area. Internal exposure from [137]Cs, as opposed to dose rates from external exposure, decreases more slowly in the general population and its contribution to total body exposure increases gradually [14]. Therefore, our study attempts to identify the potential effect of low internal doses of [137]Cs on the digestive organs of the human body, as it is known that [137]Cs accumulates in muscles and visceral organs. In Semoshkina et al's study, it was reported that [137]Cs was highly transferred to the spleen, lungs, heart, muscles, kidneys, skin and bones in horse tissue taken 90 days after the beginning of radionuclide administration [15]. We assume that a majority of the residents have similar patterns and frequencies of internal exposure from [137]Cs in the body, that continuously varies over time. This assumption is based on as previous 10-year study from the same area [13].

This study's participants were aged between 18–83 years. The number of participants was relatively well distributed. They predominantly consisted of women, whose average levels of internal radiation were lower than that of men, which is consistent with other preceding studies in the same field and area [12, 13]. The proportion of women with detectable levels of internal radiation was also lower than men, because women tend to be vigilant and tend avoid risks [16]. The average levels of internal Bq/kg and the proportion of detected with radioactivity among men were higher than those among women. However, the differences in the average internal Bq/kg in men and women were not significant ($p > 0.05$), as well as the proportion of detectable people was not significant ($p > 0.05$).

Furthermore, we examined the percentage of upper GI positive findings in men and women, that revealed that men have a higher percentage of upper GI positive diagnosis with significant differences ($p < 0.05$). Additionally, we found that men have a significantly larger average number of upper GI endoscopic diagnosis than women ($p < 0.01$). This finding is in line with previous studies that confirm that men tend to have a larger number of background digestive diseases by nature as opposed to women [7].

In this study, the average number of upper GI endoscopic diagnosis detected in one person was two for the entire population. Only 16 participants were diagnosed negative, while 434 were detected with 1 diagnosis. The rest (72%) of the population had two or more and up to 5 diagnosis per person. We inspected the prevalence of the numbers of upper GI endoscopic diagnosis in different age groups and their average levels of internal radiation in Bq/kg with intention to identify the existence of relationships between them. The proportions of the numbers of upper GI endoscopic diagnosis in different age range groups revealed a gradual increase in the proportions of higher numbers of upper GI endoscopic diagnosis, particularly the proportions of 3 and 4 diagnoses in older adults groups, though there was no significant difference (p>0.05). On the contrary, the average level of internal radiation in Bq/kg decreased as the age range of groups increased, excluding the highest age group that contained only six participants and was unlikely to be credible for comparison. The fact that older adults in the Chernobyl area have low internal exposure than young people was also confirmed in the previous study, that examined body burden for 10 years in a large number of residents in the same Zhytomyr region [13]. Considering the results shown in Fig 2, we believe that the Bq/kg has no association with the increase in the number of upper GI endoscopic diagnosis, that was also statistically insignificant. Instead, we are more likely to attribute these findings to the aging process of humans, that is in line with many studies devoted to GI systems in older adults.

Meanwhile, Cosset et al. reported that patients receiving infra-diaphragmatic high-dose irradiation therapy had developed various late radiation GI injuries: stomach and duodenum ulcers, severe gastritis, small bowel obstructions, and perforations; moreover, fewer patients developed two injuries at the same time [17]. Similarly, Kavanagh et al. found that doses of radiotherapy on the order of 45 Gy to the whole stomach are associated with late effects, primarily ulceration in the stomach and small bowels in approximately 5% to 7% of patients, respectively [18]. Dumic et al., in their study, stated that "gastrointestinal (GI) changes in the elderly are common", and "despite some GI disorders being more prevalent in the elderly" [19].

Besides, we examined the relationships between the number of upper GI endoscopic diagnosis and the proportion of people with detectable radioactivity, and the average Bq/kg. We first stratified the entire population into six groups by the number of upper GI endoscopic diagnosis and calculated the percentage of peopel with detectable radioactivity and average Bq/kg for each group. We then conducted the correlation analysis between Bq/kg and the number of upper GI endoscopic diagnosis that was 0.038 (Pearson correlation), demonstrating that there is nearly no relation (p>0.05). Additionally, we searched for other similar studies that could confirm or reject our results. However, we could not find similar studies that could provide either supporting or controversial references. There are only studies dedicated to the effects of external radiation on non-cancer diseases, reporting that the magnitude of the risk of non-cancer diseases at lower dose levels (for example, 0.5 Sv) is very uncertain [7].

It is widely known that the intake of wild forest food containing $^{137}$Cs will carry particles into the body, that then remain for a certain period and may gradually increase if accumulated in the body [20]. Consequently, if internal radiation is likely to cause a GI disorder or disease, people who frequently consume contaminated food and have internal radiation may have a higher average number of upper GI endoscopic diagnosis. Therefore, we conducted a statistical calculation to determine whether the group with detectable levels of internal radiation had a higher average number of upper GI endoscopic diagnosis than the group with undetectable levels. There is nearly no age difference between these groups. In contrast, the average age of the group with detectable radioactivity was even younger. The group with detectable levels of internal radiation had a higher average number of upper GI endoscopic diagnosis (2.1) than that in undetectable groups (1.9), and the difference between the two groups was statistically significant (p<0.001). However, the levels of Bq/kg detected in the participants does not

contribute to the increase in the number of upper GI endoscopic diagnosis. Given these results, we assumed that the intake of wild forest food might often be accompanied by alcohol intake as a celebration of successful harvests, hunting, or fishing. This, in turn, would increase the frequency of alcohol intake and consequently increase the number of upper GI endoscopic diagnosis in people with detectable levels of radioactivity will increase. To be more precise, we examined whether the wild food consumers have a higher proportion of alcohol cosumption. The Chi-square test results showed that there was a higher proportion of alcohol drinkers among those who consumed wild forest food. They showed significant differences ($p < 0.05$), suggesting that the increased number of upper GI endoscopic diagnosis in groups with detectable radioactivity are more likely to have originated due to alcohol consumption.

We conducted a regression analysis involving almost all the main characteristics of our population, that revealed that sex, intake of wild forest food, and alcohol consumption significantly affects the number of upper GI endoscopic diagnosis ($p < 0.01$). Men tend to have a higher number of upper GI endoscopic diagnosis than women, in line with the results of our sex analysis shown above ($p < 0.05$). Intake of wild forest foodstuff significantly affects the number of upper GI endoscopic diagnosis, consistent with the results shown in Table 2 of detectable and not-detectable groups ($p < 0.01$). We attribute this phenomenon to alcohol consumption, as people consuming forest and wild food tend to be alcohol drinkers, as mentioned above. We believe that the intake of wild food and its relationship with GI findings should be investigated more precisely. Expectedly, the analysis of Bq/kg and the number of upper GI endoscopic diagnosis showed that the former does not contribute to the increase in the number of upper GI endoscopic diagnosis.

Alcohol consumption clearly affected the number of upper GI endoscopic diagnosis and was statistically significant ($p < 0.001$). Studies on alcohol consumption and its effects on various organs have already been proven to cause damage to the digestive organs. Bishehsari et al. point out that alcohol-induced intestinal inflammation may be at the root of multiple organ dysfunctions and chronic disorders associated with alcohol consumption, including chronic liver disease, neurological disease, GI cancers, and inflammatory bowel syndrome [21]. Smoking, on the contrary, does not show statistical evidence that it affects the number of upper GI endoscopic diagnosis. Further, it is worth underscoring that the adjusted $R^2$ was 0.050, indicating that all independent items can together explain only a tiny portion of the increase in the number of upper GI endoscopic diagnosis.

There are some limitations in our study. First, there is no control cohort group to compare and estimate excessive risks for upper GI endoscopic diagnosis induced by internal radiation. However, we found no relationship between the internal radiation dose and number of upper GI endoscopic diagnosis in this study. Second, we did not conduct follow-ups with the participants over a longer research period or estimate their life-long accumulated low-dose radiation. Additionally, we also did not assess the effect of life-long radiation on the increase of in the number upper GI endoscopic diagnosis. However, we admit that these points should be considered and included in future studies. Third, our participants were those who sought medical assistance from the Medical Center with some symptoms or underlying disorders in their digestive systems. If we assess the number of upper GI endoscopic diagnosis in the general population, we can obtain more exact information, but may also face ethical challenges as GI endoscopy is highly invasive. Next, there may be some other confounding factors apart from those that we considered. We did not sort classifications of diseases nor examine their types, frequency, and causes. Therefore, we assessed the association of radiation with GI findings using the number of upper GI endoscopic diagnosis as an indication of GI lesions for assessing the effects. We assumed that if the GI tract is affected by the extent of chronic internal irradiation, it is likely to cause various GI injuries in each person in contaminated areas.

Our study shows that the number of upper GI endoscopic diagnosis was significantly affected by alcohol consumption. We assume that the number of upper GI endoscopic diagnosis may be an appropriate indicator of the extent of GI lesions.

In conclusion, we found that there is no obvious relationship between the increase in the number of upper GI endoscopic diagnosis and internal exposure, specifically with Bq/kg. However, there is strong evidence that alcohol consumption is more likely to affect the increase in the number of upper GI endoscopic diagnosis. There is still ambiguity regarding the effect of low-dose radiation on non-cancer diseases, particularly on the GI organs. Discussions and investigations are still ongoing in the field of low dose effects on the health of human beings. This issue requires further rigorous investigation and various research methods that can ensure more reliable evidence and underpin related hypotheses.

## Supporting information

**S1 Dataset.**
(XLSX)

**S1 Questionnaire. Inclusivity in global research checklist.**
(DOCX)

## Acknowledgments

The authors would like to thank the staff at the Zhytomyr Inter-Area Medical Diagnostic Center and all the participants who took part in this study. We would like to thank Editage (www. editage.com) for English language editing.

## Author Contributions

**Conceptualization:** Izumi Yamaguchi, Jumpei Takahashi, Alexander Gutevich, Naomi Hayashida.

**Data curation:** Yesbol Sartayev, Izumi Yamaguchi, Jumpei Takahashi, Alexander Gutevich, Naomi Hayashida.

**Formal analysis:** Yesbol Sartayev, Izumi Yamaguchi.

**Funding acquisition:** Yesbol Sartayev, Alexander Gutevich.

**Investigation:** Yesbol Sartayev, Izumi Yamaguchi, Jumpei Takahashi, Naomi Hayashida.

**Methodology:** Yesbol Sartayev, Izumi Yamaguchi, Jumpei Takahashi, Naomi Hayashida.

**Project administration:** Jumpei Takahashi.

**Resources:** Alexander Gutevich.

**Software:** Alexander Gutevich.

**Supervision:** Alexander Gutevich, Naomi Hayashida.

**Validation:** Jumpei Takahashi.

**Visualization:** Yesbol Sartayev.

**Writing – original draft:** Yesbol Sartayev.

**Writing – review & editing:** Yesbol Sartayev, Naomi Hayashida.

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
