## [Decision Letter · Decision Letter 0]

13 Jul 2022

PONE-D-22-16737Gastrointestinal findings in residents living in areas affected by the Chernobyl nuclear accidentPLOS ONE

Dear Dr. Hayashida,

Thank you for submitting your manuscript to PLOS ONE. After careful consideration, we feel that it has merit but does not fully meet PLOS ONE’s publication criteria as it currently stands. Therefore, we invite you to submit a revised version of the manuscript that addresses the points raised during the review process.

We look forward to receiving your revised manuscript.

Kind regards,

Mohamad Syazwan Mohd Sanusi

Academic Editor

PLOS ONE

Journal Requirements:

2.  Please include a complete copy of PLOS’ questionnaire on inclusivity in global research in your revised manuscript. Our policy for research in this area aims to improve transparency in the reporting of research performed outside of researchers’ own country or community. The policy applies to researchers who have travelled to a different country to conduct research, research with Indigenous populations or their lands, and research on cultural artefacts. The questionnaire can also be requested at the journal’s discretion for any other submissions, even if these conditions are not met.  Please find more information on the policy and a link to download a blank copy of the questionnaire here: https://journals.plos.org/plosone/s/best-practices-in-research-reporting. Please upload a completed version of your questionnaire as Supporting Information when you resubmit your manuscript.”

3.  "Please clarify how participants were recruited, and how they were provided information about and given access to the opt-out consent form.

4. Please provide additional details regarding participant consent. In the ethics statement in the Methods and online submission information, please ensure that you have specified what type you obtained (for instance, written or verbal, and if verbal, how it was documented and witnessed). If your study included minors, state whether you obtained consent from parents or guardians. If the need for consent was waived by the ethics committee, please include this information.

Reviewers' comments:

Reviewer's Responses to Questions

**Comments to the Author**

1. Is the manuscript technically sound, and do the data support the conclusions?

Reviewer #1: Yes

Reviewer #2: Yes

Reviewer #3: No

Reviewer #4: Partly

Reviewer #5: Yes

2. Has the statistical analysis been performed appropriately and rigorously? 

Reviewer #1: Yes

Reviewer #2: Yes

Reviewer #3: No

Reviewer #4: No

Reviewer #5: Yes

3. Have the authors made all data underlying the findings in their manuscript fully available?

Reviewer #1: Yes

Reviewer #2: Yes

Reviewer #3: Yes

Reviewer #4: Yes

Reviewer #5: Yes

4. Is the manuscript presented in an intelligible fashion and written in standard English?

Reviewer #1: Yes

Reviewer #2: Yes

Reviewer #3: No

Reviewer #4: Yes

Reviewer #5: Yes

5. Review Comments to the Author

Reviewer #1: The manuscript reports a study on influence of internal low dose radiation exposure caused by 137Cs on gastrointestinal organs of residents of Zhytomyr region living near the Chernobyl Nuclear Power Plant. Possible effect of other factors, such as alcohol or forest foodstuff consumption and smoking, on appearance of gastrointestinal findings was also examined. The paper is well written, the results are clearly presented. Interesting results arising from the research were demonstrated, which should stimulate further work in this particular field. Generally, the scientific level of the manuscript is suitable for publication in PLOS ONE journal, nevertheless there are few little points which have to be clarified before publication:

1) Fig. 3: there is no X-axis label;

2) Do the authors take into account the effect of external radiation exposure?

3) Why 137Cs was chosen? Is it possible, that 90Sr, having almost identical radioactive half-life, but totally different biological half-life, can also contribute to formation of gastrointestinal findings? Was the impact of present eating habits of people excluded?

4) Have all people in the cohort lived in the area all the time since the accident? Was the migration excluded?

Reviewer #2: Review of the manuscript entitled “Gastrointestinal findings in residents living in areas affected by the Chernobyl nuclear accident” submitted to PLOS ONE.

The authors investigated the effects of internal radiation coming from contaminated food on the gastrointestinal organs and induction of gastrointestinal diseases.

The authors investigated the health of the group people living in contaminated areas after the fallout of the Chernobyl Nuclear Power Plant and being exposed internally and externally to 137Cs several decades after the accident. The Whole-Body Counter and Gastro Endoscope were used to detect outcomes to in the study participants. They were also asked to fill questionnaire. The authors assumed that alcohol is the major cause o the increase of gastrointestinal findings, most likely accompanied by the intake of wild forest foodstuff. Other factors, like the average level of becquerel per kg of body mass and smoking, did not have significant statistical differences.

The paper is prepared correctly. The abstract gives all necessary information about the background and results. In Introduction the authors described the present knowledge regarding to the subject of the article. The results look interesting. The authors described very important problem of the health of people living in irradiated areas. The disadvantage of the manuscript is alack of the control group.

The manuscript is recommended for publication after minor correction. In results, the authors stated that about 1,620 participants took part in the study. They were exactly 1,620.

Reviewer #3: Major comments:

The study investigates possible associations between five types of gastro-intestinal (GI) disease and radioactive fallout of the Chernobyl accident, notable Cs137, which has been ingested mostly from wild food intake. The study cohort consisted of 1620 participants as patients of the Zhytomyr medical center with self reported GI findings. Patients have been asked to fill in a questionnaire concerned with lifestyle and diet habits. Internal exposure was assessed with a whole body counter (WBC) with a detection limit of 270 Bq Cs137 per body.

The study touches an important area in which data is rare and results are uncertain at best. It is therefore be welcomed. The authors provide a clear introduction into the topic and mention the limitations. Most importantly, the study subjects are self-selected with GI findings. In this case only the frequency of these findings may possibly be related to radiation.

The biological half-life of Cs137 is about 70 days. Keeping this in mind, the added value of WBC measurements for the present analysis should be more clearly explained. Counts above the detection limit may only occur, if patients have ingested contaminated food within a relatively short time period before the examimation. Hence, the long term exposure at low dose rates from ingested wild food cannot be captured with these measurments. In line with this observations, the measured WBC counts are not correlated with the number of GI findings per patient.

The results of the regression analysis are interesting but I suggest to perform additional calculations. Table 3 displays coefficients of univariate regression, which should be used to inform more involved modeling. Multivariate regression for men and women separately might be able to reveal an interaction between alcohol consumption and wild food intake, when we assume that alcohol consumption is lower in women compared to men. Alternatively an interaction term can be directly inserted in a multivariate regression model.

Minor comments:

Figure 3: “detectable” and “detected” are used in confusion. Please apply the term consistently. I suggest to use “patients with WBC counts of Cs137 above the detection limit”.

Reviewer #5: This epidemiologic study is focused on the GI diseases incidence in the selected population sample. This study attempts to investigate whether there is some effect of long-term internal low-dose radiation exposure on the GI organs causing GI diseases. This hypothesis was evaluated on a group of 1,620 participants undergoing WBC measurement, gastrointestinal examination and personal questionnaire about their lifestyle and diet habits.

It is a pity that there is no comparison with a control group, i.e. population non-exposed to Cs-137 and/or not intentionally looking for a medical examination and/or alcohol non-consumers/smokers. These facts significantly decrease the value of the study.

What was the MDA value in Bq/kg of bodyweight (e.g. 270 Bq per body vs. average 6.2±11.8 Bq/kg vs. detectable all 20.6±12.8 Bq/kg)?

I suggest to evaluate and estimate the long-term doses to the GI tract. This information would be of great general interest and needs some deeper analysis. Actually detected WBC activities are just very rough indication of possible GI damage risks. This part should be revised. Instead of correlation analysis between the age, WBC detected activity and GI findings it would be more interesting to try to estimate the integral lifetime low-doses based on actual WBC data and a questionnaire possibly monitoring subject habits, place of living and potential long-term exposure.

Data of the study were not provided due to personal data protection - however some anonymised data could be possibly provided in some shortened summarized form.

6. PLOS authors have the option to publish the peer review history of their article (what does this mean?). If published, this will include your full peer review and any attached files.

Reviewer #1: No

Reviewer #2: No

Reviewer #3: No

Reviewer #4: No

Reviewer #5: No

---

## [Author Response · Author response to Decision Letter 0]

18 Aug 2022

August 18, 2022

Dear Reviewers,

Thank you for inviting us to submit a revised draft of our manuscript entitled, ‘Gastrointestinal findings in residents living in areas affected by the Chernobyl nuclear accident’ [ms. no PONE-D-22-16737] to PLoS ONE. We also appreciate the time and effort you and each of the reviewers have dedicated to providing insightful feedback on ways to strengthen our paper. Thus, it is with great pleasure that we resubmit our article for further consideration. We have incorporated the changes that reflect the detailed suggestions you have so graciously provided. We also hope that our edits and the responses we provide below satisfactorily address all the issues and concerns you and the reviewers have noted. 

First of all, we apologize that eight minors were included in our study due to a mistake in age calculation, who do not meet the criteria for participation in our study. We excluded them from our analyses and performed all the calculations and evaluations again. However, the findings and conclusions of our study did not change and remain the same. 

To facilitate your review of our revisions, the following is a point-by-point response to the questions and comments delivered in your E-mail dated 13 July 2022. Also, we had native speakers of English proofread our English writing again.

Editor

Comment 1. Please ensure that your manuscript meets PLOS ONE’s style requirements, including those for file naming.

Response. We have ensured that the manuscript meets all the PLOS ONE’s style requirements, including those for file naming. 

Comment 2. Please include a complete copy of PLOS ONE’s ‘questionnaire on inclusivity’ in global research in your revised manuscript. Please upload a completed version of your questionnaire as Supporting Information when you submit your manuscript.

Response. We have completed the questionnaire on inclusivity and will upload it when we submit our revised manuscript as recommended. 

Comment 3. Please clarify how participants were recruited and how they were provided information about and given access to the opt-out consent. 

Response. We apologize for the lack of clarity. We re-examined all the documents issued by the University Ethics Committee and ensured that all consent forms were submitted in writing. Accordingly, we have revised the section on participant recruitment and elucidated the process of providing and receiving written informed consent. 

(Line 142-151 of clean copy)

All of them were under the medical surveillance of the Zhytomyr Inter-Area Medical Diagnostic Center (Medical Center) that provided medical services for residents from our research area. We invited all patients who sought medical assistance in the Medical Center for any GI symptoms or digestive organ disorders during the study period to participate in the study. Residency registration within the research area at the moment of examination was mandatory. Those who agreed to participate in the study, initially received a detailed description of the process and content of the study, and were then asked to provide written consent. Additionally, they filled out a questionnaire regarding their lifestyle and diet habits and underwent gastrointestinal endoscopy and measurement of their internal body burden on the Whole Body-Counter (WBC). 

(Line 185-187 of clean copy)

Informed consent was obtained from each individual through the written form, which indicated agreement for participation in the research.

Comment 4. Please provide additional details regarding participants consent. In the ethics statement in the Methods and online submission information, please ensure that you have specified what type you obtained (for instance, written or verbal, and if verbal, how it was documented and witnessed). If your study includes minors, state whether you obtained consent form parents of guardians. If the need for consent was waived by the ethics committee, please include this information.

Response. We have revised the section concerning obtaining consent and elucidated the process of providing information and receiving consent in the Methods. Please refer to line 142-151 and to line 185-187 regarding consent from minors (shown above in Comment 3). 

As for minor participants, we discovered that eight minors were included in our study due to a mistake in age calculation, who do not meet with the criteria for participation in our study. Consequently, we excluded them from our analyses and performed all the calculations and evaluations again with the remaining participants. In general, the overall results and findings of our study did not change, and the study conclusions remain the same. 

Comment 5. We note that you have indicated that data from this study are available upon request. PLOS only allows data to be available upon request if there are legal or ethical restrictions on sharing data publicly. 

a) If there are ethical or legal restrictions on sharing de-identified data set, please explain them in detail and who has imposed them. Please also provide contact information for data access. 

b) If there are no restrictions, please upload minimal anonymized data set necessary to replicate your study findings as either Supporting Information files or to a stable, public repository and provide with relevant URLs, DOIs, or accession numbers. Please see guidelines on how to de-identify and prepare clinical data for publication. 

Response. We have provided bare minimal anonymised data to enable replication and reproduction of the findings reported in our paper. 

Comment 6. We note that you have included the phrase “data not shown” in your manuscript. Unfortunately, this does not meet our data sharing requirements. PLOS does not permit reference to inaccessible data. If the data are not a core part of the research being presented in your study, we ask that you remove the phrase that refers to these data.

Response. We have removed the phrase “data not shown” and ensured that all numbers and results can be reproduced using the provided data. 

Reviewer #1:

We are grateful to reviewer #1 for the critical comments and useful suggestions that have helped us improve our paper considerably. As indicated in the responses that follow, we have considered all these comments and suggestions while revising our paper.

Comment 1. Fig.3: There is no X-axis label

Response. We have edited the part of Figure 3 that was commented on and have labelled the X-axis. 

Comment 2. Do the authors take into account the effect of external radiation exposure.

Response. Our main research target was only low-dose internal exposure from 137Cs and GI findings, and therefore, we did not account for external exposure. 

(Line 301-308 of clean copy)

 Internal exposure from 137Cs, as opposed to dose rates from external exposure, decreases more slowly in the general population and its contribution to total body exposure increases gradually [12]. Therefore, our study attempts to identify the potential effect of low internal doses of 137Cs on the digestive organs of the human body, as it is known that 137Cs accumulates in muscles and visceral organs. In Semoshkina et al’s study, it was reported that 137Cs was highly transferred to the spleen, lungs, heart, muscles, kidneys, skin and bones in horse tissue taken 90 days after the beginning of radionucllide administration [13].

Comment 3. Why 137Cs was chosen? It is possible, that 90Sr, having almost identical radioactive half-life, but totally different biological half-life, can also contribute to formation of gastrointestinal findings. Was the impact of present eating habits of people excluded. 

Response. We have amended the first paragraph of the Discussion section, where we elucidate our reason for choosing 137Cs as the primary interest of our study, as opposed to another radionuclides.

(Line 296-308 of clean copy)

In this study, the key focus was internal exposure emitted from 137Cs and its potential effects on the gastrointestinal system of the human body. A majority of studies concerning the CNPP accident that investigate the effects of radiation on the human body have concentrated their primary interests in 137Cs and internal exposure. 137Cs has a greater effect on people than other radionuclides due to its properties, such as a long half-life, amount released into environment and dispersion in a wider area. Internal exposure from 137Cs, as opposed to dose rates from external exposure, decreases more slowly in the general population and its contribution to total body exposure increases gradually [12]. Therefore, our study attempts to identify the potential effect of low internal doses of 137Cs on the digestive organs of the human body, as it is known that 137Cs accumulates in muscles and visceral organs. In Semoshkina et al’s study, it was reported that 137Cs was highly transferred to the spleen, lungs, heart, muscles, kidneys, skin and bones in horse tissue taken 90 days after the beginning of radionucllide administration [13].

The impact of eating habits was not excluded in our study. 

(Line 308-311 of clean copy)

We assume that a majority of the residents have simillar patterns and frequencies of internal exposure from 137Cs in the body, which continuously varies over time. This assumption is based on as earlier 10-year study from the same area [14].

Comment 4. Have all people in the cohort lived in the area all the time since the accident? Was the migration excluded.

Response. We added some additional information confirming that all participants only needed residency registration within the study area at the time of examination. 

(Line 144-147 of clean copy)

We invited all patients who sought medical assistance in the Medical Center for any GI symptoms or digestive organ disorders during the study period to participate in the study. Residency registration within the research area at the moment of examination was mandatory.

Reviewer #2:

We are grateful to reviewer #2 for the critical comments and useful suggestions that have helped us improve our paper considerably. As indicated in the responses that follow, we have considered all these comments and suggestions while revising our paper.

Reviewer #3:

We are grateful to reviewer #2 for the critical comments and useful suggestions that have helped us improve our paper considerably. As indicated in the responses that follow, we have considered all these comments and suggestions while revising our paper.

Comment 1. I suggest performing additional calculations. Multivariate regression for men and women separately might be able to reveal an interaction between alcohol consumption and wild food intake, when we assume that alcohol consumption is lower in women compared to men. 

Response. We have performed the suggested analysis and added its results to the Results section.

 (Line 289-294 of clean copy)

We also performed regression analysis for women separately, as it could reveal an interaction between alcohol consumption and wild food intake, assuming that alcohol consumption is lower in women compared to men. Regression analysis for the only-women group revealed results similar to Table 3, indicating significance for age, alcohol consumption, intake of wild food (p<0.05), while smoking and the level of Bq/kg seemingly did not affect the number of GI findings (p>0.05).

Comment 2. Figure 3: “detectable” and “detected” are used in confusion. Please apply the term consistently. I suggest using “patients with WBC counts of Cs137 above the detection limit”.

Response.

We have edited the specified sentence in accordance with your suggestion as shown in line 257-258 “participants with 137Cs exposure above the WBC’s detection limit”. We have also revised “detectable” to “detected” in Figure 3 for consistency. 

(Line 262-263 of clean copy)

Fig 3 demonstrates the average Bq/kg and percentage of participants with 137Cs exposure above the WBC’s detection limit for each number of GI findings in the group.

Reviewer #5:

We are grateful to reviewer #5 for the critical comments and useful suggestions that have helped us improve our paper considerably. As indicated in the responses that follow, we have considered all these comments and suggestions while revising our paper.

Comment 1. It is pity that there is no comparison with a control group, i.e. population non-exposed to Cs-137 and/or not intentionally looking for a medical examination and/or alcohol non-consumers/smokers. These facts significantly decrease the value of the study. 

Response. We agree with your comment and have accordingly mentioned these imperfections in the limitations. 

Comment 2. What was the MDA value in Bq/kg of bodyweight (e.g. 270 Bq per body vs average 6.2�11.8 Bq/kg vs. detectable all 20.6 Bq/kg).

Response. The MDA value of the Whole-Body Counter we used during our study was 270 Ber per body. The average Bq/kg for the entire population was 6.2�11.8 whereas the average Bq/kg for the group with exposure above the Whole-Body Counter’s detection limit was 20.6. We have additionally edited the titles of rows in Table 2 for clarity. 

Comment 3. I suggest evaluating and estimating the long-term doses to the GI tract. Instead of correlation analysis between the age, WBC detected activity and GI findings, it would be more interesting to try to estimate the integral lifetime low-dose based on WBC data and a questionnaire, possibly monitoring subject habits, place of living and potential long-term exposure. 

Response.

We agree that long-term doses and lifetime exposure would be of greatly interest to research. However, our study is not designed to follow-up with affected individuals and monitor participant habits and place of living. We have added these points to the limitations. 

(Line 429-433 of clean copy)

We did not conduct follow-ups with the participants any longer research period or estimate their life-long accumulated low-dose radiation. Additionally, we also did not assess the effect of life-long radiation on the increase of GI findings. However, we admit that these points should be considered in future studies and included in research methods.

Comment 4. Data of the study were not provided due to personal data protection – however some anonymised data could be possibly provided in some shortened summarized form.

Response.

We have provided anonymised data to enable replication and reproduction of the study findings. 

With the incorporation of these amendments to our final manuscript, we hereby resubmit our manuscript for a secondary evaluation. We hope that the revised version of our paper is now suitable for publication in PLoS ONE, and we look forward to hearing from you at your earliest convenience.

Sincerely,

Naomi Hayashida, M.D., PhD. 

Professor of Division of Strategic Collaborative Research, Center for Promotion of Collaborative Research on Radiation and Environment Health Effects, Atomic Bomb Disease Institute, Nagasaki University

1-12-4 Sakamoto, Nagasaki 852-8523, JAPAN

TEL: +81-95-819-8507

FAX: +81-95-819-8508

E-mail: naomin@nagasaki-u.ac.jp

---

## [Decision Letter · Decision Letter 1]

6 Sep 2022

PONE-D-22-16737R1Gastrointestinal findings in residents living in areas affected by the Chernobyl nuclear accidentPLOS ONE

Dear Dr. Hayashida,

Thank you for submitting your manuscript to PLOS ONE. After careful consideration, we feel that it has merit but does not fully meet PLOS ONE’s publication criteria as it currently stands. Therefore, we invite you to submit a revised version of the manuscript that addresses the points raised during the review process.

We look forward to receiving your revised manuscript.

Kind regards,

Mohamad Syazwan Mohd Sanusi

Academic Editor

PLOS ONE

Journal Requirements:

Reviewers' comments:

Reviewer's Responses to Questions

**Comments to the Author**

1. If the authors have adequately addressed your comments raised in a previous round of review and you feel that this manuscript is now acceptable for publication, you may indicate that here to bypass the “Comments to the Author” section, enter your conflict of interest statement in the “Confidential to Editor” section, and submit your "Accept" recommendation.

Reviewer #2: All comments have been addressed

Reviewer #4: (No Response)

2. Is the manuscript technically sound, and do the data support the conclusions?

Reviewer #2: Yes

Reviewer #4: Yes

3. Has the statistical analysis been performed appropriately and rigorously? 

Reviewer #2: Yes

Reviewer #4: Yes

4. Have the authors made all data underlying the findings in their manuscript fully available?

Reviewer #2: Yes

Reviewer #4: Yes

5. Is the manuscript presented in an intelligible fashion and written in standard English?

Reviewer #2: Yes

Reviewer #4: Yes

6. Review Comments to the Author

Reviewer #2: The authors have introduced modifications according to the suggestions of the reviewers.

Above corrections have enriched the content of manuscript and have made it better. In my opinion the manuscript in the present form is suitable for publication.

Reviewer #4: My comments have not been addressed but I suggested minor revisions. I recommended that the authors should extend their regression analysis. Has this been done?

7. PLOS authors have the option to publish the peer review history of their article (what does this mean?). If published, this will include your full peer review and any attached files.

Reviewer #2: No

Reviewer #4: No

---

## [Author Response · Author response to Decision Letter 1]

7 Sep 2022

Dear Reviewer #4,

Thank you for inviting us to submit a revised draft of our manuscript entitled, ‘Gastrointestinal findings in residents living in areas affected by the Chernobyl nuclear accident’ [ms. no PONE-D-22-16737R1] to PLoS ONE. We also appreciate the time and effort you have dedicated to providing insightful feedback on ways to strengthen our paper. Thus, it is with great pleasure that we resubmit our article for further consideration. We have incorporated the changes that reflect the detailed suggestions you have so graciously provided. We also hope that our edits and the responses we provide below satisfactorily address all the issues and concerns you have noted. 

Comment 1. I recommended that the authors should extend their regression analysis. Has this been done?

Response. We have performed the suggested analysis and added its results to the Results section. 

(Line 289-294 of clean copy)

We also performed regression analysis for women separately, as it could reveal an interaction between alcohol consumption and wild food intake, assuming that alcohol consumption is lower in women compared to men. Regression analysis for the only-women group revealed results similar to Table 3, indicating significance for age, alcohol consumption, intake of wild food (p<0.05), while smoking and the level of Bq/kg seemingly did not affect the number of GI findings (p>0.05).

With the incorporation of these amendments to our final manuscript, we hereby resubmit our manuscript for a secondary evaluation. We hope that the revised version of our paper is now suitable for publication in PLoS ONE, and we look forward to hearing from you at your earliest convenience.

---

## [Editor Report · Decision Letter 2]

29 Sep 2022

PONE-D-22-16737R2Gastrointestinal findings in residents living in areas affected by the Chernobyl nuclear accidentPLOS ONE

Dear Dr. Naomi,

Thank you for submitting your manuscript to PLOS ONE. After careful consideration, we feel that it has merit but does not fully meet PLOS ONE’s publication criteria as it currently stands. Therefore, we invite you to submit a revised version of the manuscript that addresses the points raised during the review process. Please submit your revised manuscript by 28 September 2022. If you will need more time than this to complete your revisions, please reply to this message or contact the journal office at plosone@plos.org. Please include the following items when submitting your revised manuscript:A rebuttal letter that responds to each point raised by the academic editor and reviewer(s). You should upload this letter as a separate file labeled 'Response to Reviewers'.A marked-up copy of your manuscript that highlights changes made to the original version. You should upload this as a separate file labeled 'Revised Manuscript with Track Changes'.An unmarked version of your revised paper without tracked changes. You should upload this as a separate file labeled 'Manuscript'.If applicable, we recommend that you deposit your laboratory protocols in protocols.io to enhance the reproducibility of your results. Protocols.io assigns your protocol its own identifier (DOI) so that it can be cited independently in the future. For instructions see: https://journals.plos.org/plosone/s/submission-guidelines#loc-laboratory-protocols. Additionally, PLOS ONE offers an option for publishing peer-reviewed Lab Protocol articles, which describe protocols hosted on protocols.io. Read more information on sharing protocols at https://plos.org/protocols?utm_medium=editorial-email&utm_source=authorletters&utm_campaign=protocols.

We look forward to receiving your revised manuscript.

Kind regards,

Mohamad Syazwan Mohd Sanusi

Academic Editor

PLOS ONE

Journal Requirements:

Academic editor: Congrats to the author, it is a great work. I enjoy reading your work. Kindly find my comment below.

Title- The title is confusing. The “Gastrointestinal finding” looks like a hanging title. Gastrointestinal cancer incidence? Gastrointestinal diseases? Gastrointestinal symptoms? Please revise the title, it must be concise and reflecting your work, not too general.

Please define the GI findings. Are they non-cancer disease? Lien 145-146 give a glace of definition GI finding, but I believe it is useful in abstract and introduction.

Abst, Line 24-25 – There is no concrete evident that all the locals are digesting the contaminated Cs-137 food stuffs. The author should be careful with the statement unless there is a report from WHO, ICRP epidemiological studies that show the significant evidence from important Cs biomarkers eg faeces and bone etc. The first line 23-24 also should be revised. Externally exposures from gammas from Cs-137 are true but not for internal exposures of the all the local peoples considering biological half lives of Cs-137 in few tenth days.

Abst, methodology – Please reorganise the method and approaches used in this study. The abstract writing is very important component to highlight background, problem statement, methodology, result and discussion and mostly importantly the conclusion. In your case, the methods and approach are not in a good order.

- You can just combine and make it concise. All the data on internal 137Cs concentration and GI findings in people were collected from 2016 to 2018 in the Zhytomyr region, Ukraine. The Whole-Body Counter and Gastro Endoscope were used to detect outcomes in the study participants.

- What are your main data inputs? Cs-137 and GI symptoms? Each data inputs have parameters of score/merits to see the correlation? eg. intake of wild forest food, smoking habits, and alcohol consumption.

- How many main data collected and how many secondary data (questionnaires) are obtained? How many participates in WBC and GI endoscope?

Abst, result/discussion – limited results are given. The scientific values are important but not included in abstract? Range and average internal accumulated Bq are detected in GI system. Any finding from gastro endoscope? You mentioned the alcohol is a major cause for GI finding, yet no statistical hypothesis test to assess the likelihood of alcohol impacts on GI findings using analysis of variance, or the non-parametric test eg Welch’s anova or Kruskal Wallis test etc. All the parameters that affecting the GI finding eg. sex, intake of wild forest foodstuff, and alcohol consumption need to be assessed too. Line 175-180 gives the details, but found none in abstract

a) -However, the proportion of a higher number of GI findings increased with the age of the groups. This tendency can be seen in the proportions of the 3 and 4 GI findings. However, the proportion of 5 GI findings was detected only in two age groups, namely 51–60 years and 61–70 years. We also assessed the significance of the average Bq/kg in different age groups, which turned out to have no significant difference (p=0.461). The highets average Bq/kg was observed in the group of 81-83 years old. However this group may not be representative, as it only containes 6 participants. If this group is excluded, the younger the age, the higher the level of internal radiation exposure.

b) Factors such as Bq/kg and smoking did not affect the increase of number of GI findings.

c) age, sex, intake of wild food, and alcohol consumption contribute significantly to the increase in the number of GI findings.

d) They showed significant differences (p<0.05), suggesting that the increased levels of GI findings in detectable groups are more likely to have originated due to alcohol consumption.

The highlighted inputs above should be included in abstract.

Abst, results – “Other factors, like the average level of a becquerel per kilogram of body mass (Bq/kg) and smoking”. Kindly revised the statement. Its confusing. What activity is this? Cs-137? The for the smoking what is the tangible data ? how do you compare that?

Abst, conclusion – no conclusion have been drawn.

Abst, recommendation - My opinions is that you draw a conclusion from your 2 years of work before suggesting a study that efficient in proving you hypothesis from this study.

Intro, line 55-60 – instead of stating the existence of BEIR VI’s comprehensive report on effect off low dose, it is more appropriate if the author highlighting the BEIR, IAEA, UNSCEAR and WHO findings on Cs-137 contaminated foodstuffs in Chernobyl.

Intro, 61-65 – “highest exposed recovery workers”. Kindly check the suitability to include this information because it may be a dominated by major gamma external exposures. The written introductions are lack of focus. The author highlights the “recovery workers” which out of the topic according to this work aims of motivation and title. It is necessary up to these points, the authors only addressing the literature work of investigation of Cs-137/other isotopes intakes by the Chernobyl’s publics.

Intro, 66-75 –the whole paragraph just highlighting the finding for the thyroid cases from ingested iodine contaminated food. Can you make some hypothesis of these finding in relation to your work GI findings based for Cs-137 contamination? At least a line stating that increased I-131 thyroid cases after few years of the incident shows that high possibility of unavoidable Cs-137 ingestion through local foodstuffs.

Line 76-116 –These para(s) highlighting the BEIR VI findings of non-cancer risks based on LSS & abs data from UNSCEAR reports in your work. Discussing the background is necessary however you include broaden scopes of internal organ non-cancer incidences. This manuscript will be impactful to readers if you narrow the scopes by focusing the literature works/background studies on GI impacts from low dose or Cs-137 ingestions.

Line 124-126 - The statement need citations.

Line 163-164 - We used the number of GI findings detected in each participant to indicate the extent of gastrointestinal lesions. What number? A score? How do you scale it? 1 until 5?..please elaborate after the line.

Line 163 – Please elaborate the criteria for finding 1 – 5? Extent of lesion and condition?

Line 166 – radiation? Or activity levels?

Line 170 – cleaned? Filtered?

Methodology – Please include the work flowcharts of your work. From fill form, wbc screening – GI endoscopy, classification 0 Bq or higher, data treatment, statistical tes etc.

Table 1 - P-value of what significance test? Men vs women?

Table 1 – Define body burden and detected subject

Result – Based on highlighted lines below, kindly explain how do you treat the difference sizes of population groups in the Mann Whitney U test? How to assure no bias output from the analysis? P=0.461 value may be affected by the significant difference of group size?

We also assessed the significance of the average Bq/kg in different age groups, which turned out to have no significant difference (p=0.461). The highets average Bq/kg was observed in the group of 81-83 years old.

Line 279 - dependent variables? Not independent?

Line 296-326 – redundant as intro and methodology. Please remove. Discussion is only to discuss your inputs and highlight the finding which started at line 327.

---

## [Author Response · Author response to Decision Letter 2]

7 Nov 2022

Dear Dr. Sanusi, Academic Editor

PLOS ONE

Thank you for inviting us to resubmit a revised draft of our manuscript entitled, ‘Gastrointestinal findings in residents living in areas affected by the Chernobyl nuclear accident’ [PONE-D-22-16737R2] to PLoS ONE. 

We appreciate your earnest comments and valuable suggestions that have notably improved the structure and alleviated the comprehension of our manuscript.

We have incorporated the changes in the manuscript to reflect the detailed suggestions you have so graciously provided. We also hope that our edits and the responses we provide below satisfactorily address all the issues and concerns. 

To facilitate your review of our revisions, below are point-by-point responses to the questions and comments delivered via your E-mail dated September 30, 2022. 

We have considered all the comments and suggestions that were submitted and responded as provided below.

Comment 1. 

Response. 

We have ensured that the reference list in the manuscript is complete and correct. We did not find a record of retraction for any paper we have included in the reference list and our manuscript. 

Comment 2. 

Title- The title is confusing. The “Gastrointestinal finding” looks like a hanging title. Gastrointestinal cancer incidence? Gastrointestinal diseases? Gastrointestinal symptoms? Please revise the title, it must be concise and reflecting your work, not too general.

Response. 

We changed the title of our paper to “The association between upper gastrointestinal endoscopic findings and internal radiation exposure in residents living in areas affected by the Chernobyl nuclear accident”.

Comment 3-1. 

Please define the GI findings. Are they non-cancer disease? 

Response. 

The GI findings we investigated in this study included all types of upper GI endoscopic diagnoses that can be diagnosed by a gastroenterologist according to the International Classification of Disease. We use the word “diagnosis” to distinguish from “findings.” “Upper GI endoscopic diagnosis” represents each finding such as gastritis, duodenitis, duodenogastric reflux, gastroesophageal reflux, etc.

Comment 3-2.

Line 145-146 give a glace of definition GI finding, but I believe it is useful in abstract and introduction.

Response. 

We added details regarding the definition of GI findings in the introduction. Adding more details to the abstract seems difficult as the journal puts limitation of 300 words. We use the word “diagnosis” to distinguish from “findings.” “Upper GI endoscopic diagnosis” represents each finding such as gastritis, duodenitis, duodenogastric reflux, gastroesophageal reflux, etc.

We amended Introduction section according to your recommendation. 

(Line 121-126 of clean copy)

The findings of upper GI endoscopic examination were interpreted according to the International Classification of Diseases (ICD) of World Health Organization (WHO). Common upper GI endoscopic diagnoses such as gastritis, duodenitis, duodenogastric reflux, gastroesophageal reflux, stomach ulcer, diaphragmatic hernia, and other diagnoses, including a few cases of cancer, were diagnosed in a wide range of combinations among most of the screened participants. 

Comment 4. 

Abstract, Line 24-25 – There is no concrete evident that all the locals are digesting the contaminated Cs-137 food stuffs. The author should be careful with the statement unless there is a report from WHO, ICRP epidemiological studies that show the significant evidence from important Cs biomarkers eg faeces and bone etc. The first line 23-24 also should be revised. Externally exposures from gammas from Cs-137 are true but not for internal exposures of the all the local peoples considering biological half-lives of Cs-137 in few tenth days. 

Response. 

We revised corresponding sentences according to your recommendation. 

(Line 22-26 of clean copy)

Many people living around the Chernobyl Nuclear Power Plant (CNPP) have been exposed to 137Cs for several decades after the CNPP accident. Although half-life of 137Cs is about 30 years, some wild forest foodstuffs are contaminated by 137Cs even now. We pointed out in a previous report that low-dose internal radiation has been occasionally detected in people’s body. 

Comment 5. 

Abst, methodology – Please reorganise the method and approaches used in this study. The abstract writing is very important component to highlight background, problem statement, methodology, result and discussion and mostly importantly the conclusion. In your case, the methods and approach are not in a good order. 

- You can just combine and make it concise. All the data on internal 137Cs concentration and GI findings in people were collected from 2016 to 2018 in the Zhytomyr region, Ukraine. The Whole-Body Counter and Gastro Endoscope were used to detect outcomes in the study participants. 

- What are your main data inputs? Cs-137 and GI symptoms? Each data inputs have parameters of score/merits to see the correlation? eg. intake of wild forest food, smoking habits, and alcohol consumption.

- How many main data collected and how many secondary data (questionnaires) are obtained? How many participates in WBC and GI endoscope? 

Response. 

We added and modified the order of the sentences for more clarity and consistency. As the guidelines of PLOS ONE for the abstract are substantially limited, we amended the Abstract concisely and elaborated Materials and methods and Results sections of the manuscript according to your suggestions. 

(Line 30-38 of clean copy)

Overall, 1,612 residents were assessed for internal 137Cs concentration using Whole-Body Counter and their digestive organs were screened with upper GI endoscopy from 2016-2018 in the Zhytomyr region, Ukraine. All participants answered to the questionnaire including their background, intake of wild forest foodstuff, intake frequency, smoking habits, and alcohol consumption. We checked the number of upper GI endoscopic diagnosis per person to assess the extent of damage to the upper digestive organs. Next, we statistically analyzed associations between this number and age, sex, level of internal exposure dose, alcohol consumption, wild forest foodstuff intake, and smoking.

Comment 6. 

Abst, result/discussion – limited results are given. The scientific values are important but not included in abstract? Range and average internal accumulated Bq are detected in GI system. Any finding from gastro endoscope? You mentioned the alcohol is a major cause for GI finding, yet no statistical hypothesis test to assess the likelihood of alcohol impacts on GI findings using analysis of variance, or the non-parametric test eg Welch’s anova or Kruskal Wallis test etc. All the parameters that affecting the GI finding eg. sex, intake of wild forest foodstuff, and alcohol consumption need to be assessed too. Line 175-180 gives the details, but found none in abstract

a) -However, the proportion of a higher number of GI findings increased with the age of the groups. This tendency can be seen in the proportions of the 3 and 4 GI findings. However, the proportion of 5 GI findings was detected only in two age groups, namely 51–60 years and 61–70 years. We also assessed the significance of the average Bq/kg in different age groups, which turned out to have no significant difference (p=0.461). The highets average Bq/kg was observed in the group of 81-83 years old. However this group may not be representative, as it only containes 6 participants. If this group is excluded, the younger the age, the higher the level of internal radiation exposure. 

b) Factors such as Bq/kg and smoking did not affect the increase of number of GI findings.

c) age, sex, intake of wild food, and alcohol consumption contribute significantly to the increase in the number of GI findings.

d) They showed significant differences (p<0.05), suggesting that the increased levels of GI findings in detectable groups are more likely to have originated due to alcohol consumption.

The highlighted inputs above should be included in abstract.

Response. 

We reflected all the recommendations provided above in the abstract of our manuscript concisely. Though the guidelines of PLOS ONE for abstract put some limitations on the number of words and structure, as follows: 

• Describe the main objective(s) of the study

• Explain how the study was done, including any model organisms used, without methodological detail

• Summarize the most important results and their significance

• Not exceed 300 words

(Line 38-44 of clean copy)

Consequently, we revealed that the number of GI diagnosis is significantly increased by factors such as sex, intake of wild forest foodstuff, and alcohol consumption. However, the average level of internal exposure of 137Cs and smoking did not relate to the number of GI diagnosis. Thus, the results of multiple regression revealed that alcohol consumption is independently related to the number of GI diagnosis that is most likely accompanied by the intake of wild forest foodstuff. In conclusion, the low-dose internal exposure may not affect the digestive organs of residents living around CNPP.

Comment 7. 

Abst, results – “Other factors, like the average level of a becquerel per kilogram of body mass (Bq/kg) and smoking”. Kindly revised the statement. Its confusing. What activity is this? Cs-137? The for the smoking what is the tangible data? how do you compare that?

Response. 

We added further details for clarity and to facilitate the comprehension of readers. 

(Line 40-41 of clean copy)

However, the average level of internal exposure of 137Cs and smoking did not relate to the number of GI diagnosis.

Comment 8. Abst, conclusion – no conclusion have been drawn.

Response. 

According to your suggestion, we added the conclusion in the abstract.

(Line 41-44 of clean copy)

Thus, the results of multiple regression revealed that alcohol consumption is independently related to the number of GI diagnosis that is most likely accompanied by the intake of wild forest foodstuff. In conclusion, the low-dose internal exposure may not affect the digestive organs of residents living around CNPP.

Comment 9. 

Abst, recommendation - My opinions is that you draw a conclusion from your 2 years of work before suggesting a study that efficient in proving you hypothesis from this study.

Response. 

We slightly changed the structure of the sentence and believe that it now conveys the meaning better. There are limitations stated in the guidelines of PLOS ONE for the abstract regarding the number of words and conclusion.

(Line 43-44 of clean copy)

In conclusion, the low-dose internal exposure may not affect the digestive organs of residents living around CNPP.

Comment 10. 

Intro, line 55-60 – instead of stating the existence of BEIR VI’s comprehensive report on effect off low dose, it is more appropriate if the author highlighting the BEIR, IAEA, UNSCEAR and WHO findings on Cs-137 contaminated foodstuffs in Chernobyl. 

Response. 

Contaminated foodstuff intake was not the main focus of this study. This study estimated the effects of chronic low internal radiation on the human body’s organs, particularly on the digestive organs, which remains unexplored. Many papers have reported that it may cause some health effects after several decades have passed. Therefore, we needed to cite recent findings regarding low-dose health outcomes reported by prominent reports that are related to our study. We supplemented information with reference to earlier studies that investigated internal exposure in bodies of people from contaminated areas. 

(Line 106-116 of clean copy)

A few studies investigating the presence of radioactivity in the bodies of people living in contaminated areas around CNPP, showed that a substantial percentage of the population—almost 50% in the beginning of the study year—had some level of radiation [12, 13]. A study conducted from 1996 to 2008 found that 513 participants or 0.35% of the study population had an annual internal radiation dose exceeding 1 mSv, which is the dose limit set by the International Commission on Radiological Protection for the general public [12]. Despite our study was conducted from 2009 to 2018, on screening the residents around CNPP for internal radiation, we found fewer residents (53 participants, 0.02%) with higher levels of dose and radiation detected in their bodies. Consequently, there is still uncertainty regarding the effects of chronic low-dose internal radiation and its health outcomes [13].

Comment 11. 

Intro, 61-65 – “highest exposed recovery workers”. Kindly check the suitability to include this information because it may be a dominated by major gamma external exposures. The written introductions are lack of focus. The author highlights the “recovery workers” which out of the topic according to this work aims of motivation and title. It is necessary up to these points, the authors only addressing the literature work of investigation of Cs-137/other isotopes intakes by the Chernobyl’s publics.

Response. 

We removed the part related to the high-exposed recovery workers but retained the part that states the final finding of the BEIR report. 

Comment 12. 

Intro, 66-75 –the whole paragraph just highlighting the finding for the thyroid cases from ingested iodine contaminated food. Can you make some hypothesis of these finding in relation to your work GI findings based for Cs-137 contamination? At least a line stating that increased I-131 thyroid cases after few years of the incident shows that high possibility of unavoidable Cs-137 ingestion through local foodstuffs.

Response. 

We added this paragraph to provide the general overview and outcomes caused by internal radiation exposure on the health of people living in contaminated areas around Chernobyl. In respond to your comments, we also deleted partially from lines 68 to 73.

Comment 13. 

Line 76-116 –These para(s) highlighting the BEIR VI findings of non-cancer risks based on LSS & abs data from UNSCEAR reports in your work. Discussing the background is necessary however you include broaden scopes of internal organ non-cancer incidences. This manuscript will be impactful to readers if you narrow the scopes by focusing the literature works/background studies on GI impacts from low dose or Cs-137 ingestions.

Response. 

We deleted Lines 82–86 and 94–101 as they were less relevant to our study; however, the other paragraphs are related to the topic of our study, since most of the upper GI findings are of non-cancer diseases.

Comment 14. 

Line 124-126 - The statement needs citations.

Response. 

These mentioned suspicions that internal radiation causes GI diseases were expressed verbally by the doctors from the Medical Centre, while we worked on another joint research with them. They pointed out that people from contaminated areas seemingly have relatively higher incidence of GI disease than people in other areas according to their observation and suggested investigating whether the presence of internal radiation possibly causes an increase in GI findings. There is no specific paper to cite. 

Comment 15. 

Line 163-164 - We used the number of GI findings detected in each participant to indicate the extent of gastrointestinal lesions. What number? A score? How do you scale it? 1 until 5? please elaborate after the line.

Response. 

We changed the mentioned sentence as follows and added more details for better comprehension. 

(Line 166-170 of clean copy)

Upper GI endoscopic findings in participants were diagnosed by professional gastroenterologists according to the ICD of WHO. We used the number of upper GI endoscopic diagnosis per person to indicate the extent of GI damages. The number of upper GI endoscopic diagnosis detected in one participant in our dataset varied from 0 to 5 diagnoses.

Comment 16. 

Line 163 – Please elaborate the criteria for finding 1 – 5? Extent of lesion and condition?

Response. 

We changed the mentioned sentence as follows and added more details to improve comprehension. 

(Line 166-172 of clean copy)

Upper GI endoscopic findings in participants were diagnosed by professional gastroenterologists according to the ICD of WHO. We used the number of upper GI endoscopic diagnosis per person to indicate the extent of GI damages. The number of upper GI endoscopic diagnosis detected in one participant in our dataset varied from 0 to 5 diagnoses. Each participant could have various combinations of upper GI diseases, such as gastritis, duodenitis, duodenogastric reflux, gastroesophageal reflux, stomach ulcer, diaphragmatic hernia, and so on.

Comment 17. 

Line 166 – radiation? Or activity levels?

Response. 

We added the word “radioactivity” for clarity. 

(Line 163-164 of clean copy)

The minimum detectable radioactivity level of 137Cs on this WBC was 270 Bq per body.

Comment 18. 

Line 170 – cleaned? Filtered? 

Response. 

We supplemented that all the data was cleaned and filtered, as shown below.

(Line 181-183 of clean copy)

All the data was cleaned, filtered and grouped by certain characteristics, such as sex, age, number of detected upper GI endoscopic diagnosis, wild forest food and alcohol intake, and smoking habits.

Comment 19. 

Methodology – Please include the work flowcharts of your work. From fill form, wbc screening – GI endoscopy, classification 0 Bq or higher, data treatment, statistical test etc.

Response. 

We amended the sentences to include details of the workflow to enable simple understanding. 

(Line 144-159 of clean copy)

We invited all patients who sought medical assistance in the Medical Center for any upper GI symptoms or digestive organ disorders that required GI endoscopic intervention during the study period. Residency registration within the research area at the moment of examination was mandatory. Those who agreed to participate in the study, initially received a detailed description of the process and content of the study and were then asked to provide written consent. Afterward, they first completed a questionnaire regarding their lifestyle and dietary habits. Questionnaires were distributed in hard copies and prepared in the Russian language. It consisted of four pages in A4 size and included the respondent’s name, address, date of birth, informed consent, milk and forest food intake, alcohol consumption, smoking habits and their frequency. Once they finished filling the questionnaire, they were invited to udergo upper GI endoscopy and measurement of their internal body burden on the Whole-Body Counter (WBC). All the data collected from the questionnaires, GI endoscopic examinations, and WBC measurements were then used to assess their effects of internal exposure on the upper GI endoscopic findings and identify their associations and contributions. 

(Line 185-192 of clean copy)

When the internal exposure of the participants was below detectable levels, they were considered and qualified as “0 Bq.” The relevant and necessary statistical tests were conducted and represented in the appropriate way. All statistical analyses were performed on IBM SPSS Statistics 25.0 software. The Mann-Whitney U test and Chi-square tests were used for statistical significance and the determination of averages and proportions. We also ran correlation tests and univariate regression analysis to test the contributions of several variables. P-values lower than 0.05 were considered significant.

Comment 20. 

Table 1 - P-value of what significance test? Men vs women?

Response. 

The P-value in Table 1 represents the significance in the values for men and women. We added an annotation under Table 1.

(Line 222 of clean copy)

P-value represents the significance in values of men and women.

Comment 21. 

Table 2 – Define body burden and detected subject

Response. 

We deleted the term “body burden” from this paragraph as it may cause some confusion in understanding it; nonetheless, the meanings of body burden and internal exposure are the same throughout our paper. We added the following as annotations under Table 2.

(Line 232 of clean copy)

Body burden is the value of radioactivity in participants represented in Bq/kg.

Comment 22. 

– Based on highlighted lines below, kindly explain how do you treat the difference sizes of population groups in the Mann Whitney U test? How to assure no bias output from the analysis? P=0.461 value may be affected by the significant difference of group size?

We also assessed the significance of the average Bq/kg in different age groups, which turned out to have no significant difference (p=0.461). The highets average Bq/kg was observed in the group of 81-83 years old.

Response. 

For the analysis mentioned above, we used Tamhane test, which tests the mean (Avg.Bq/kg in each group) difference in significancy compared to other groups. 

The distribution of the dataset for all age categories, avg. Bq/kg, and standard deviation are presented in the following table. The group with the fewest participants is the 80-years-old age group, which consisted of only 6 persons. Other groups were distributed relatively evenly. Although the 10-year age group has 53 participants, the number of participants is large enough. The number of participants in each age-group is sufficient to conclude that there is no bias (except in the 80-years group). Additionally, considering their standard deviation, they do not fluctuate into a large extent, which, to us, means that the data are not biased. We believe that the data of avg.Bq/kg. for age groups is fair and not biased (except the 80-years group). 

 For more clarity, we added more details into the mentioned paragraph about the number of participants in the groups and the test used for identification of the significance of average Bq/kg between groups.

(Line 272-278 of clean copy)

We also assessed the significance of the average Bq/kg between age groups using ANOVA test. The ANOVA multiple Comparisons Tamhane test showed no significant difference between age groups (p=0.461). The highest average Bq/kg was observed in the group of 81-83 years old. However, this group may not be representative, as it only contains 6 participants, while all other groups consist of 53-380 participants. If the age group of 81-83 years is excluded, then the younger the age, the higher the level of internal radiation exposure.

Comment 23. 

Line 279 - dependent variables? Not independent?

Response. 

Your remark is correct; thus, we revised the sentence accordingly. 

(Line 306-307 of clean copy)

This analysis evaluates whether these independent variables affect the increase in the number of upper GI endoscopic diagnosis in individuals.

Comment 24. 

Line 296-326 – redundant as intro and methodology. Please remove. Discussion is only to discuss your inputs and highlight the finding which started at line 327.

Response. 

We find the first paragraph necessary for gaining a general understanding of our paper, and we are certain that it is relevant to our study’s topic. Nevertheless, we decided to remove the second paragraph, lines 317-331. 

With the incorporation of these amendments to our final manuscript, we hereby resubmit our manuscript for a third evaluation. We hope that the revised version of our paper is now suitable for publication in PLoS ONE, and we look forward to hearing from you at your earliest convenience.

Sincerely,

Naomi Hayashida, M.D., PhD. 

Professor of Division of Strategic Collaborative Research, Center for Promotion of Collaborative Research on Radiation and Environment Health Effects, Atomic Bomb Disease Institute, Nagasaki University

1-12-4 Sakamoto, Nagasaki 852-8523, JAPAN

TEL: +81-95-819-8507

FAX: +81-95-819-8508

E-mail: naomin@nagasaki-u.ac.jp

---

## [Editor Report · Decision Letter 3]

16 Nov 2022

The association between upper gastrointestinal endoscopic findings and internal radiation exposure in residents living in areas affected by the Chernobyl nuclear accident

PONE-D-22-16737R3

Dear Prof. Dr. Naomi,

We’re pleased to inform you that your manuscript has been judged scientifically suitable for publication and will be formally accepted for publication once it meets all outstanding technical requirements.

Kind regards,

Mohamad Syazwan Mohd Sanusi

Academic Editor

PLOS ONE

Additional Editor Comments (optional):

Congrats to the authors for the hardwork and impacful research.
---

## [Editor Report · Acceptance letter]

18 Nov 2022

PONE-D-22-16737R3 

The association between upper gastrointestinal endoscopic findings and internal radiation exposure in residents living in areas affected by the Chernobyl nuclear accident 

Dear Dr. Hayashida:

I'm pleased to inform you that your manuscript has been deemed suitable for publication in PLOS ONE. Congratulations! Your manuscript is now with our production department. 

Kind regards, 

on behalf of

Dr. Mohamad Syazwan Mohd Sanusi 

Academic Editor

PLOS ONE